# 5-HT2AR and NMDAR psychedelics induce similar hyper-synchronous states in the rat cognitive-limbic cortex-basal ganglia system

Ivani Brys[1,2,3], Sebastian A. Barrientos [1,3], Jon Ezra Ward[1], Jonathan Wallander[1], Per Petersson[1,3] & Pär Halje [1✉]

The profound changes in perception and cognition induced by psychedelic drugs are thought to act on several levels, including increased glutamatergic activity, altered functional connectivity and an aberrant increase in high-frequency oscillations. To bridge these different levels of observation, we have here performed large-scale multi-structure recordings in freely behaving rats treated with 5-HT2AR psychedelics (LSD, DOI) and NMDAR psychedelics (ketamine, PCP). While interneurons and principal cells showed disparate firing rate modulations for the two classes of psychedelics, the local field potentials revealed a shared pattern of synchronized high-frequency oscillations in the ventral striatum and several cortical areas. Remarkably, the phase differences between structures were close to zero, corresponding to <1 ms delays. Likely, this hypersynchrony has major effects on the integration of information across neuronal systems and we propose that it is a key contributor to changes in perception and cognition during psychedelic drug use. Potentially, similar mechanisms could induce hallucinations and delusions in psychotic disorders and would constitute promising targets for new antipsychotic treatments.

---

[1] The Group for Integrative Neurophysiology and Neurotechnology, Department of Experimental Medical Science, Lund University, Lund, Sweden. [2] Research Group in Neuroscience and Experimental Psychology, Federal University of Vale do São Francisco, Petrolina, Brazil. [3] Department of Integrative Medical Biology, Umeå University, Umeå, Sweden. ✉email: par.halje@med.lu.se

Converging evidence show that psychedelics are effective at treating several neuropsychiatric conditions and that their therapeutic effect depends mainly on their ability to induce neuroplasticity[1–3]. Less is known about how psychedelics alter brain activity to induce the acute changes in perception and cognition that they are most known for[4]. The acute psychedelic state is important to study in a medical context, both for its potential contribution to long-term therapeutic effects but also as a model for psychosis. More fundamentally, psychedelics-induced changes in brain activity might reveal processes important for the study of consciousness[5].

Psychedelic drugs are primarily classified phenomenologically based on their ability to induce a psychedelic experience. Nevertheless, it is well established that classic psychedelics like lysergic acid diethylamide (LSD) and 2,5-dimethoxy-4-iodo-amphetamine (DOI) exert their effects mainly through their agonistic action on 5-HT2A receptors[6–8], and consequently this class of psychedelics are often referred to as serotonergic or 5-HT2AR psychedelics. However, dissociative anesthetics, such as ketamine and phencyclidine (PCP), produce psychedelic-like experiences primarily via non-competitive antagonism of N-methyl-D-aspartate (NMDA) glutamate receptors[9,10] Here, we will refer to this class of drugs as NMDAR psychedelics. Despite the different primary action of 5-HT2AR and NMDAR psychedelics, increased glutamatergic neurotransmission in cortico-limbic networks has been proposed as a common downstream effect linked to the psychedelic state[4,11,12]. In particular, the psychedelic state has been linked to glutamate-dependent depolarizing membrane currents in a subpopulation of pyramidal cells in medial prefrontal cortex (mPFC)[8,13,14]. However, this increased glutamate signaling does not result in general network excitation and the effect on neuronal spiking activity is complex. The very few investigations performed in awake animals to date indicate that the highly selective 5-HT2A agonist DOI decreases spiking activity in the orbitofrontal cortex, anterior cingulate cortex and motor cortex of rodents[15,16]. In contrast, the NMDA antagonists ketamine, PCP and MK801 promote net excitation in rat mPFC, even if many individual cells also respond with inhibition[15,17–21].

Investigations of synchronized neuronal activity, in the form of local field potentials, have more consistently found overlapping effects for both 5-HT2AR and NMDAR psychedelics. Ketamine has been found to induce aberrant high-frequency oscillations (HFO, 110–180 Hz) in several corticolimbic structures in rodents[22–26]. Similarly, 5-HT2A agonists induce HFOs in the ventral striatum and mPFC of rodents[27,28]. HFOs occur to a lesser degree in normal conditions in some of these structures and are believed to enable the integration of otherwise isolated neuronal information through synchronous activity[29–31]. The olfactory bulb has been identified as an important source of psychedelics-induced HFOs[32,33]. However, local infusion experiments suggest that HFO generation can be initiated independently in multiple structures and then spread to other regions[34,35].

Systems-level analyses of the acute effects of these substances are to date limited to human imaging studies. Interestingly, a large body of evidence[36–39] have indicated that a main effect may be related to alterations in brain-wide activity and connectivity patterns, suggesting high-resolution animal studies need to be expanded to cover larger multi-structure networks to help bridging cellular findings to systems-level physiology.

For these reasons, we have here used an exceptional electrode design to investigate spike activity and HFOs in recordings from several brain structures in parallel in freely behaving rats. This allowed us to identify both local and system-wide neuronal activity changes specific to the psychedelic state, and to link

spiking activity of principal cells and interneurons to HFOs in multiple regions for the first time. We compared 5-HT2AR psychedelics (LSD, DOI) and NMDAR psychedelics (ketamine, PCP) to a non-psychedelic psychoactive control (amphetamine) and found that aberrant HFOs were consistently and specifically present during the psychedelic state in ventral striatum and in medial prefrontal, orbitofrontal and olfactory cortices. Moreover, we found that HFOs were phase synchronized between brain structures in a way that could severely influence the integration of information across neuronal systems. Hence, we propose that such hypersynchrony is a mechanism by which an altered state of consciousness can be induced, either during a psychedelic experience or as part of a psychotic episode.

## Results

**No single behavior was specific to both classes of psychedelics.** There are known behavioral cues in rodents that predict psychedelic effects in humans[40,41]. However, those cues are different for 5-HT2AR and NMDAR psychedelics; 5-HT2A agonists mainly induce spontaneous head-twitch, while NMDA antagonists mainly induce hyperlocomotion and ataxia. Therefore, to confirm that the experimental setting used (involving tethered recordings in an open-field) would not affect behaviors, and to directly assess links between brain activity and displayed behavior, the first step in our study was to compare the effects of 5-HT2AR and NMDAR psychedelics to the non-psychedelic control on a broad set of behaviors and look for psychedelic-specific behavioral changes.

As previously reported, the NMDA antagonists induced hyperlocomotion and ataxia, while 5-HT2A agonists induced head-twitch responses and amphetamine induced hyperlocomotion and stereotypy ($p < 0.001$; Fig. 1b, c, S1 and S2). In the more detailed analyses of drug-specific behavioral features, we could establish an increase in stereotypic behaviors following amphetamine treatment. Thus, characterizing the behaviors, alone, we could confirm several drug-specific behaviors but found no evidence for a behavior that was similarly altered by both 5-HT2A agonists and NMDA antagonists. Importantly, this means that the psychedelic-specific changes in neuronal activity reported below cannot be explained trivially by changes in the concomitantly displayed motor behavior.

**Firing rate modulations were similar for NMDAR psychedelics and amphetamine, but not for 5-HT2AR psychedelics.** In total, 365 units were identified from 169 recording sites in 9 freely behaving animals (Fig. 2a-c). When comparing the activity of all neurons from all structures to baseline, we observed that the spontaneous firing rates of a large proportion of cells on this global scale were affected by the drugs: 65% were inhibited and only 8% were excited with 5-HT2A agonists, while a more balanced response was found for the NMDA antagonists (40% were inhibited and 40% were excited at the $p < 0.01$ level; Figs. 2d, e). As a comparison, in response to amphetamine, 32% were inhibited and 46% were excited (Fig. 2f). Corresponding plots showing each drug individually are shown in Fig. S3.

We further investigated the specific effects of each drug on different cell populations. After grouping of the recording sites based on structural and functional similarity (Table S1), 7 groups had enough cells to warrant further analysis of population modulation (≥4 cells from both 5-HT2A and NMDA experiments; Table S2); olfactory cortex, orbitofrontal cortex, medial prefrontal cortex, temporal association area, sensorimotor cortex, ventral striatum, and integrative thalamus. Cell classification based on waveform features was performed on all these structures except thalamus, allowing for a putative division into principal

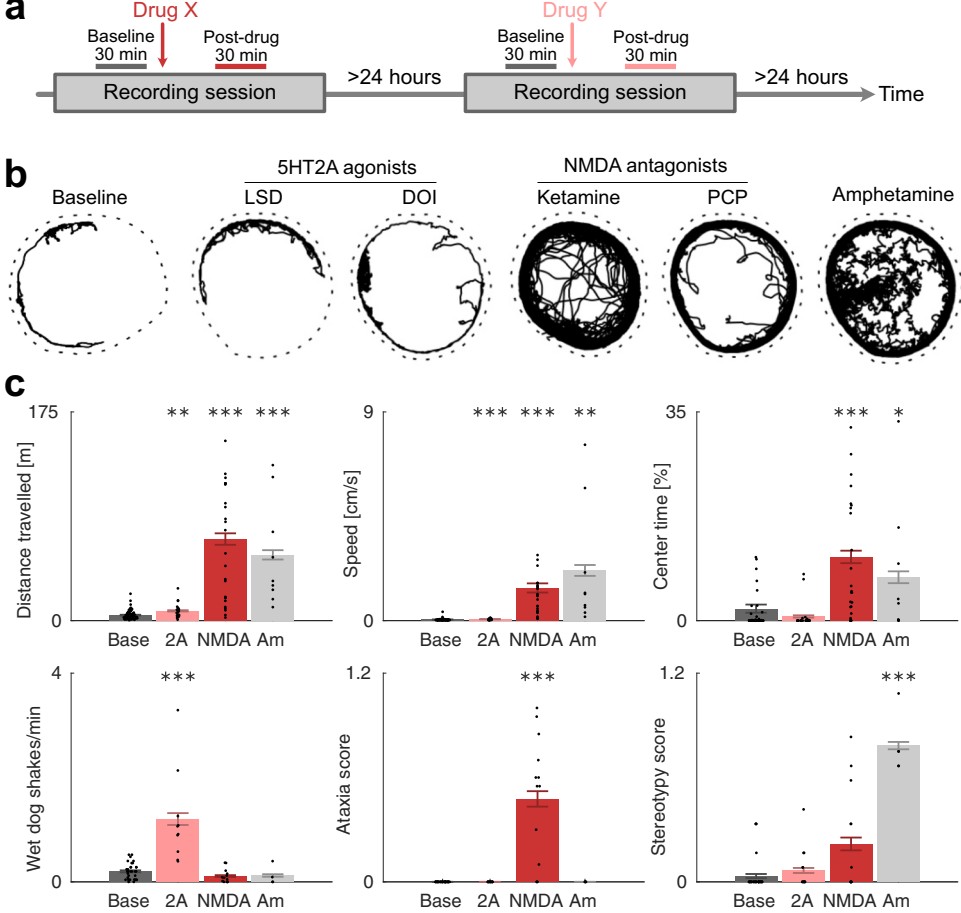

**Fig. 1 A specific pattern of behavioral changes is induced by each drug class. a** Timeline of experiment. Each recording session consisted of 60 min baseline followed by a drug injection and recording for another 60–120 min. Behavioral and electrophysiological data were averaged over -35 to -5 min for baseline measurements and 30 to 60 minutes for on-drug measurements (relative to drug injection). At least 24 h passed between recording sessions. **b** Examples of tracked motion for each condition. On baseline, the animal was mostly passive and moved occasionally in bouts along the walls of the circular arena (indicated by the dashed line). On the 5-HT2AR psychedelics LSD and DOI, the locomotion behavior was very similar to baseline. In contrast, the NMDAR psychedelics ketamine and PCP induced clear hyperlocomotion, and especially ketamine induced ataxic, unstable gait. Amphetamine induced strong hyperlocomotion and vigorous sniffing (seen here as wiggly traces). **c** Average changes in behavior for each condition (Base = baseline, 2A = LSD or DOI, NMDA = ketamine or PCP, Am = amphetamine). Bars show mean and SEM, asterisks show significance at the $p < 0.05$ (*), $p < 0.01$ (**), and $p < 0.001$ (***) levels (nested ANOVA). Top left: distance traveled during 30 min. Top center: speed. Top right: percentage of time spent in the center. Bottom left: number of head-twitch responses per minute. Bottom center: ataxia score (3 is max). Bottom right: stereotypy score (3 is max).

cells (PC), interneurons (IN) and an intermediate group of cells that could not reliably be assigned to either group (unidentified cells; X) (see Fig. S4).

For the 5-HT2A agonists the dominating effect on firing rates was inhibition in each of these structures, both in terms of net standardized rates and number of significantly up/down modulated cells (see Fig. 2g, h, left panels). For the NMDA antagonists the effect on firing rates was more mixed, with many populations responding with both inhibition and excitation (see Fig. 2h, middle panel). However, in terms of net standardized rates, we observed that INs were excited and PCs were inhibited in most structures (see Fig. 2g, middle panel). Corresponding plots showing each drug individually are shown in Figure S5.

To look for rate modulations specific to the psychedelic state, we grouped all types of psychedelic drugs and compared the firing rates to the non-psychedelic control compound amphetamine. Only neurons in integrative thalamus showed a significant difference in this comparison, with a modest reduction of −0.3 (z-scored) in the psychedelic state compared to an increase of 1.6 in the amphetamine state ($p < 0.05$). Further, the rate modulations for 5-HT2A and NMDA were not significantly different in

integrative thalamus (we also controlled that the baselines for all drugs were statistically similar; $p > 0.05$). However, on a systems level the NMDA modulations throughout all the recorded structures resembled the amphetamine modulations more than the 5-HT2A modulations. This is illustrated by Fig. 2i where IN modulation is plotted against PC modulation for each structure. NMDA and amphetamine had a similar pattern of simultaneous IN excitation and PC inhibition, while 5-HT2A had simultaneous inhibition in both IN and PC. The relative dissimilarity of the clusters was quantified by measuring their Bhattacharyya distance[42]: 5HT2A-NMDA was the most dissimilar pair with a distance of 6.6, while NMDA-amphetamine was the most similar pair with a distance of 0.43. The distance for the 5HT2A-amphetamine pair was 0.98.

In summary, 5-HT2A agonists caused widely distributed inhibition in both principal cells and interneurons. NMDA antagonists caused similar inhibitory effects in principal cells, but opposite, excitatory effects in interneurons. A psychedelic-specific modulation was found in integrative thalamus, but on a systems level the effects of the NMDA antagonists resembled the effects of amphetamine more than those of the 5-HT2A agonists.

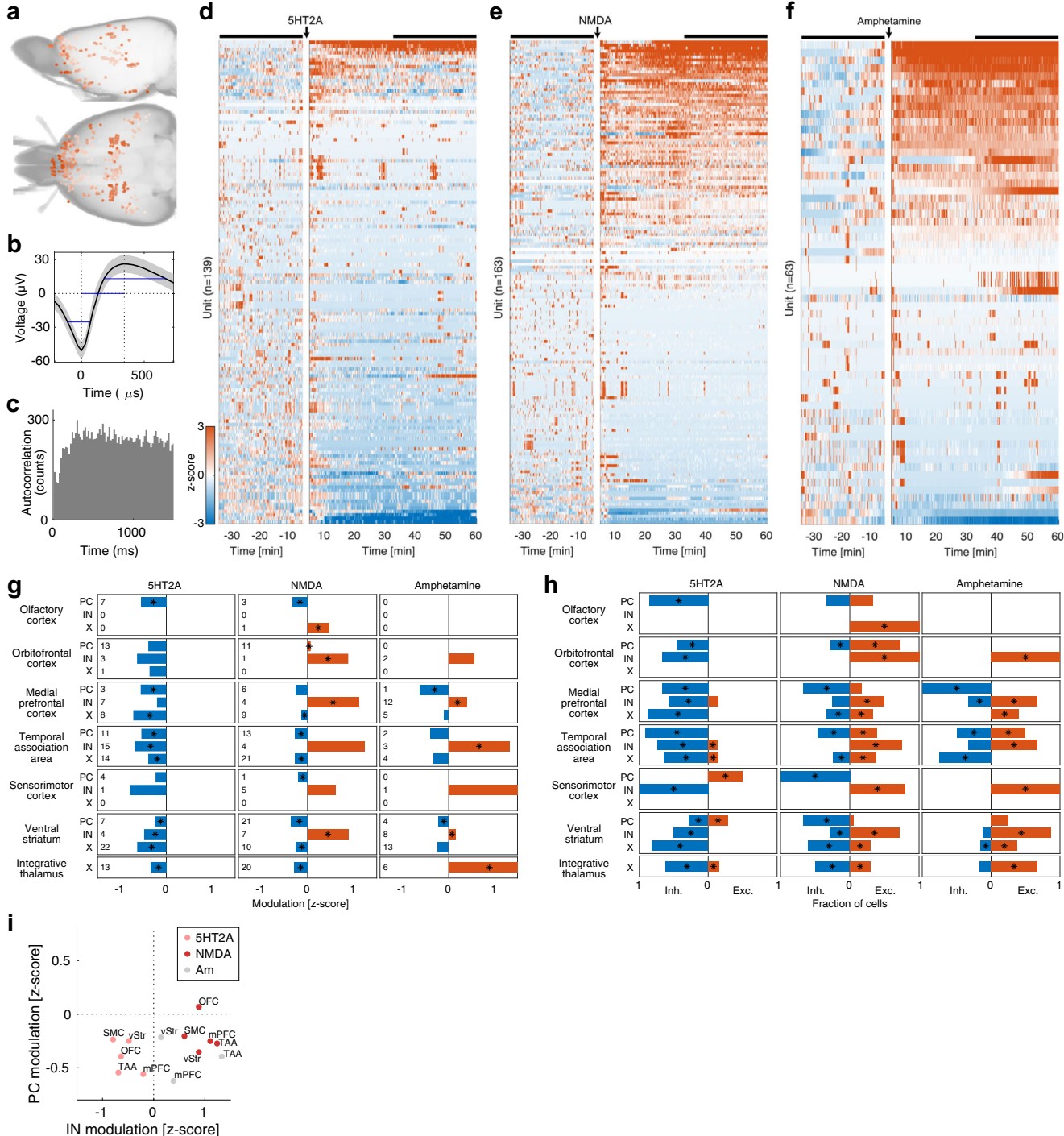

**Fig. 2 Modulation of neuronal firing rates in response to 5-HT2AR and NMDAR psychedelics. a** Electrode locations as determined by micro computed tomography. **b** Example average waveform from a single unit in prelimbic cortex. The gray area indicates SEM. The blue lines indicate waveform features (peak width, peak-to-valley time and valley width) used to classify the neuron as a putative principal cell. **c** Spike autocorrelogram of the example unit in (**b**). **d–f**. Standardized neuronal firing rate responses to 5-HT2A agonists (**d**), NMDA antagonists (**e**) and amphetamine (**f**). Each row shows the activity of a single unit and rows are rank ordered according to the response during the drug period (indicated by the black bar). **g** Average standardized neuronal firing rate responses to 5-HT2A agonists (left), NMDA antagonists (middle) and amphetamine (right) for different cell populations (PC = putative principal cells, IN = putative interneurons, X = unclassified cells). The response is calculated as average z-scores during 30 to 60 min post-drug injection compared to baseline (−35 to −5 min). Asterisks indicate significance at the $p < 0.05$ level (nested ANOVA). The numbers next to the cell labels indicate the number of cells in each population. **h** Fraction of modulated cells as response to 5-HT2A agonists (left), NMDA antagonists (middle) and amphetamine (right) for different cell populations (PC = putative principal cells, IN = putative interneurons, X = unclassified cells). The fractions of downmodulated cells are shown in blue and upmodulated cells are shown in red. Asterisks indicate significance at the $p < 0.05$ level (binomial test). **i**. Comparison of IN vs PC rate modulations reveals a similar pattern of modulation for NMDA and amphetamine (IN inhibition, PC excitation), while 5-HT2A induces inhibition in both IN and PC populations. Pink = 5-HT2A, red = NMDA, gray = amphetamine. OFC orbitofrontal cortex, mPFC medial prefrontal cortex, SMC sensorimotor cortex, TAA temporal association area, vStr ventral striatum.

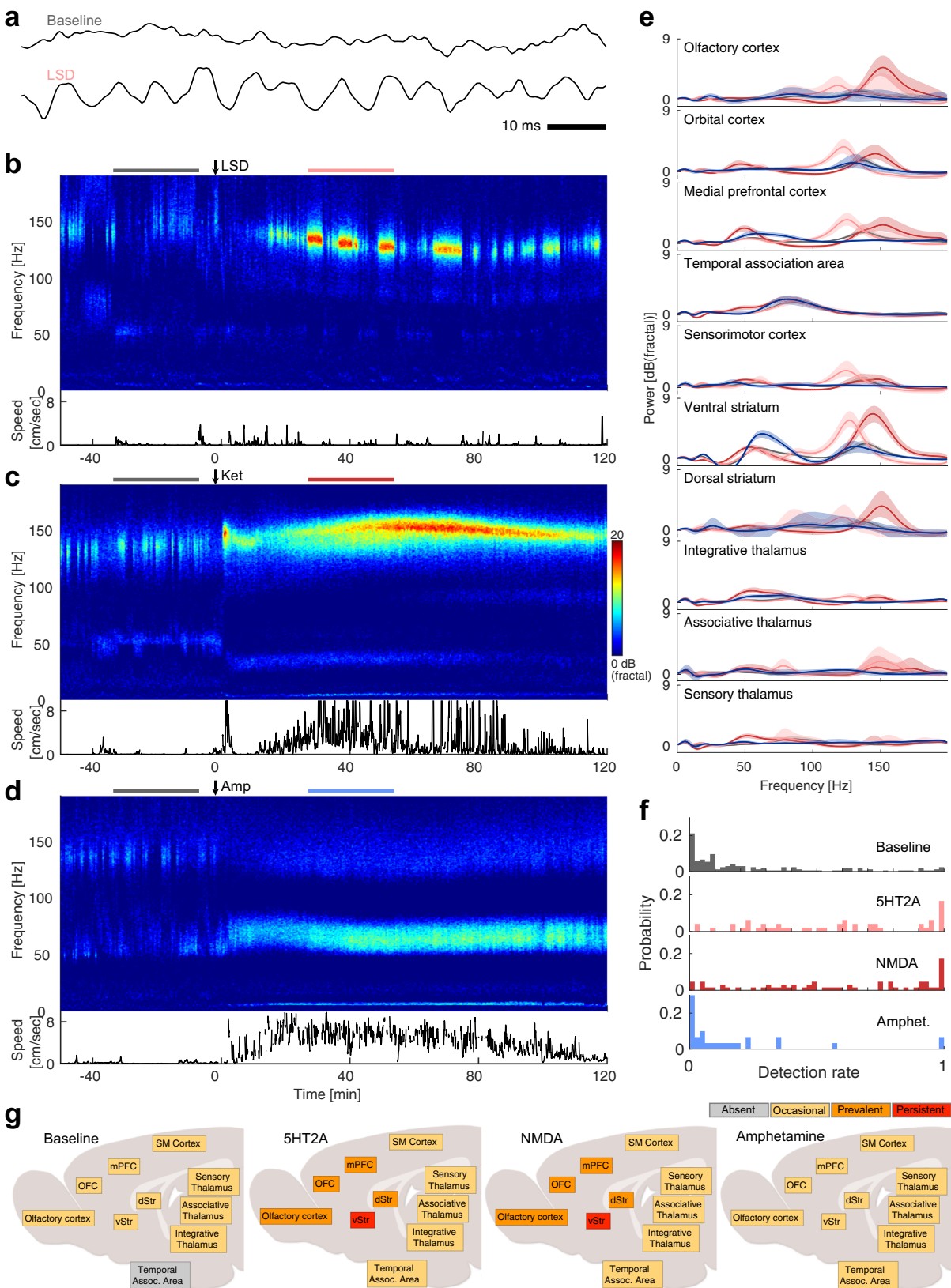

**Both classes of psychedelics induced a similar pattern of high-frequency oscillations**. To extend on previous results on psychedelics-induced HFOs, we recorded local field potentials simultaneously from several cortical and subcortical regions and characterized the extent of HFOs. As expected, both 5-HT2A agonists and NMDA antagonists consistently caused strong increases in HFOs around 150 Hz, as exemplified in Fig. 3a–d. HFOs increased both in terms of their amplitude (Fig. 3e and Fig. S6) and their prevalence (as measured by the detection rate; Fig. 3f, Figs. S7, S8). In contrast, amphetamine increased broad gamma activity (30–80 Hz) without causing HFOs (Fig. 3d–g and Fig. S8).

**Fig. 3 HFOs are enhanced by 5-HT2AR and NMDAR psychedelics but not by amphetamine. a** Example monopolar traces of LFPs recorded in the ventral striatum (shell of nucleus accumbens) before (top) and after (bottom) LSD administration. The scale bar indicates 10 ms. The signal was low pass filtered at 500 Hz. **b–d**. LFP spectrograms from bipolar ventral striatum recordings during administration of LSD (**b**), ketamine (**c**) and amphetamine (**d**). A clear increase in high-frequency oscillations around 150 Hz (HFOs) is evident after injection of ketamine or LSD, but not after amphetamine. The color scale is in units of dB$_{fractal}$, i.e. decibels normalized to the fractal noise background. The translational movement speed is shown below each spectrogram. **e** Power spectra from bipolar LFPs averaged over time and treatment groups (gray = baseline, pink = 5-HT2A agonists, red = NMDA antagonist, blue = amphetamine). The time periods used were −35 to −5 min for baseline and 30– 60 min for drug treatment relative to injection (also indicated in (**b–d**) as horizontal bars above each spectrogram). Shaded areas show bootstrapped 95% confidence intervals. **f** Distribution of HFO detection rates in different structures, showing that high detection rates are much more common in 5-HT2A and NMDA conditions. Each value is the average detection rate for a condition in a structure during one recording session. **g** Summary of HFO detection rates in different structures, showing a similar pattern of HFO prevalence for 5-HT2A agonists and NMDA antagonists. HFOs were classified as persistent (red; more than 90% detections in more than 33% of sessions), prevalent (orange; more than 50% detections in more than 33% of sessions), occasional (yellow, if not in any other class) or absent (gray; more than 5% detections in less than 5% of sessions). OFC orbitofrontal cortex, mPFC medial prefrontal cortex, SM Cortex sensorimotor cortex, vStr ventral striatum, dStr dorsal striatum.

In data pooled from all structures and individuals there was a clear shift towards higher HFO detection rates in 5-HT2A and NMDA conditions, but there was also a large spread in the detection rate distributions (Fig. 3f), indicating variation across structures. To further illustrate the extent and prevalence of HFOs in different structures we defined 4 prevalence classes based on the distribution of detection rates (see Fig. S9); for each structure and condition, HFOs were classified as persistent (>90% detections in >33% of sessions), prevalent (>50% detections in >33% of sessions), occasional (if not in any other class) or absent (>5% detections in <5% of sessions). According to this classification, 5-HT2A agonists and NMDA antagonists increased HFO prevalence in an identical pattern, while the pattern for the non-psychedelic compound amphetamine was very similar to the baseline pattern (Fig. 3g).

The main difference between the 5-HT2A and NMDA conditions was seen in the oscillation frequency. It was significantly lower for 5-HT2A agonists compared to NMDA antagonists (127±10 Hz and 143±13 Hz, respectively; $p = 0.0027$). For 5-HT2A agonists, the oscillation frequency was also significantly reduced compared to the frequency of the sporadic HFOs that occurred during baseline at 137±13 Hz ($p = 0.0140$).

**Entrainment of neuronal spiking and HFOs.** Inferring how changes in cellular firing patterns may relate to synchronized population phenomena, such as HFOs, is challenging for complex networks consisting of arrays of interconnected excitatory and inhibitory neurons. Nevertheless, as a first step towards this aim, we next characterized the relationship between psychedelics-induced HFOs and neuronal spiking by investigating if spiking occurred more frequently at certain HFO phases than others. Entrainment to the HFO was common, with 37% of cells showing entrainment during the psychedelic state (28% during baseline; entrainment was defined as $p < 0.001$ with a Rayleigh test).

An example of an entrained IN in mPFC is shown in Fig. 4a–f. In this example, entrainment was very sparse, with no clear periodicity in the autocorrelation function (Fig. 4b). The spike-triggered average showed a shift from entrainment with a slow, negative LFP deflection during baseline to entrainment with the HFO during ketamine treatment. Phase histograms revealed that the neuron fired preferentially about 2.6 radians before the trough of the HFO (corresponding to 2.7 ms; Fig. 4f).

This entrainment pattern was representative for all neurons. Histograms of the preferred phase angle showed a clear preference for phases ~2 radians before the HFO trough in PCs, INs and unclassified cells (PC: $p = 0.002$, IN: $p = 0.017$, X: $p < 0.001$, Rayleigh test; Fig. 4g). However, there was only a modest increase of 17% in the entrainment strength (mean kappa) during the psychedelic state compared to baseline

($p = 0.034$, nested ANOVA). We found no significant difference between the 5-HT2A and NMDA conditions in terms of entrainment strength ($p = 0.53$, nested ANOVA).

Entrained neurons were found in most regions, but the magnitude of entrainment differed between structures (Table S3). The PC population of ventral striatum had the strongest entrainment with a mean kappa of 0.27. Surprisingly, PCs in the temporal association area were also strongly entrained (kappa = 0.24), despite that this region showed only modest HFO amplitudes in bipolar measurements.

**Psychedelics-induced HFOs were phase-synchronized between ventral striatum and several cortical areas.** Next, we addressed the question if psychedelics-induced HFOs are generated in a single structure and propagate to other regions, or whether they are generated in a more distributed fashion. We did this by examining how HFOs in different structures were related in terms of their amplitude, frequency and phase.

Qualitatively, the HFOs had a spindle-like amplitude modulation and strong co-modulation within structures, while co-modulation between structures seemed to be weaker and more complex (Fig. 5a). The autocorrelograms of the instantaneous amplitudes were consistent with a typical spindle length of about 50 ms (the autocorrelation function had a smooth peak with FWHM = 50±18 ms; Fig. 5b). In some electrodes, the autocorrelograms revealed periodic amplitude modulation at the theta frequency (also seen in Fig. 5b). Cross-correlograms of the instantaneous amplitudes confirmed that the amplitude modulation was generally more strongly correlated within structures than between structures (median correlation coefficient 0.82±0.10 and 0.40±0.15, respectively; $p < 0.001$, Wilcoxon rank sum; Fig. 5c). However, some pairs of structures showed co-modulations that were similar in strength to the within-structure values (see the group above 0.7 in Fig. 5c). These strong amplitude correlations were found between olfactory cortex, ventral striatum and orbitofrontal cortex, as well as between medial prefrontal cortex and ventral striatum.

The oscillation frequency was remarkably similar in different brain structures when comparing simultaneous oscillations at different sites, despite large variations between individuals and between conditions (see Fig. 5d and Fig. S10). Such frequency co-modulation could occur if a single dominant source of oscillating firing rates entrains all other structures via synaptic transmission. This may be considered a likely scenario given previous data showing the importance of the olfactory bulb in the generation of HFOs[32,33]. However, further analyses of phase relationships did not support this hypothesis: Fig. 5e shows a single HFO spindle recorded simultaneously from 7 electrodes in the olfactory bulb, the ventral striatum and the orbitofrontal cortex. In this example,

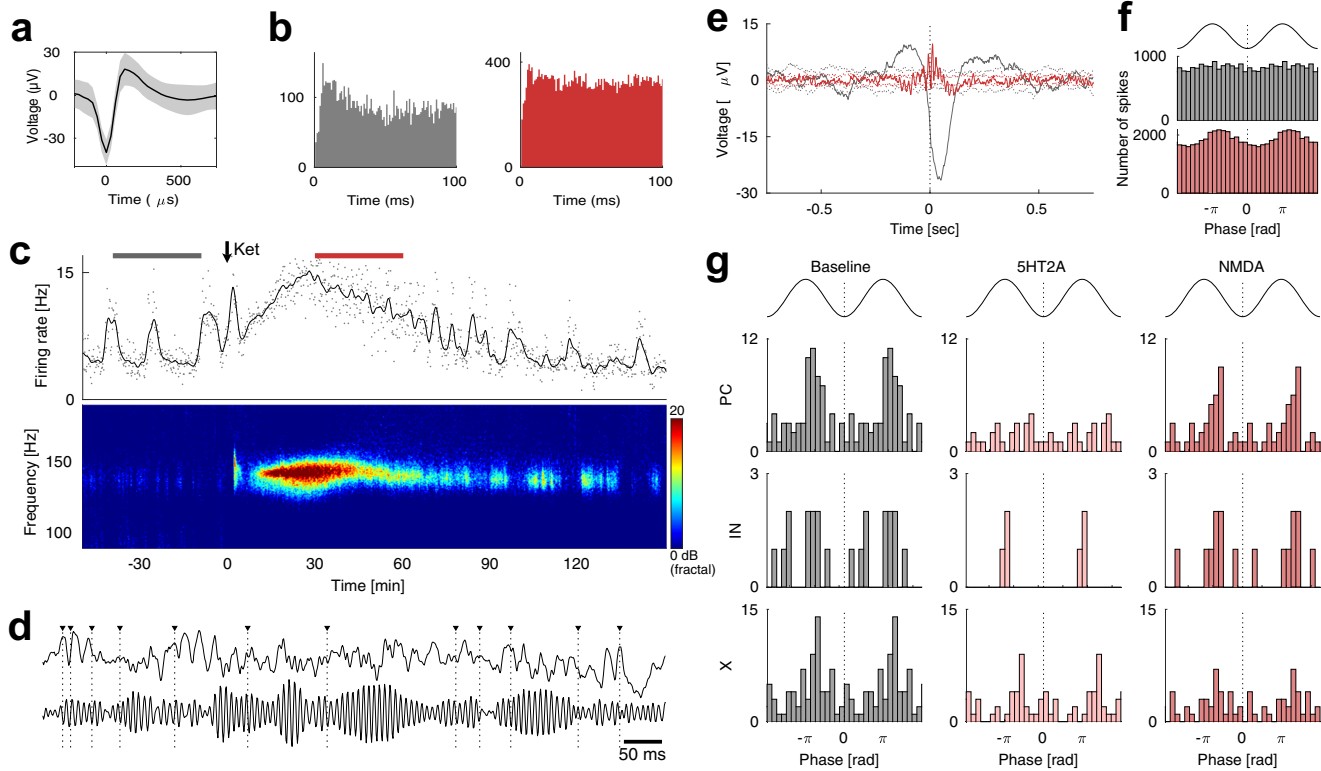

**Fig. 4 Spike entrainment to HFOs. a** Waveform of example interneuron from PFC. **b** Autocorrelograms of the example neuron during baseline (left) and during ketamine treatment (right). **c** Time-course of the firing rate of the example neuron (top) and LFP spectrogram (bottom). **d** LFP traces (top: unfiltered, bottom: bandpass filtered 110–190 Hz) with spikes from the example neuron superimposed (vertical dotted lines). **e** Spike-triggered averages for the example neuron during baseline (gray) and ketamine treatment (red). Dotted lines show 95% confidence intervals (calculated with random spike dithering). **f** LFP phase histograms of the example neuron during baseline (gray, top) and ketamine treatment (red, bottom) calculated from LFPs bandpassed at the HFO frequency (110–190 Hz). Zero is defined as the trough of the HFO. **g**. Summary histograms of the preferred phase of all entrained neurons as estimated with the von Mises distribution. All neurons with kappa>0.1 were included. Entrained neurons showed a clear preference for phases ~2 radians before the HFO trough in PCs, INs and unclassified cells (PC: $p = 0.002$, IN: $p = 0.017$, X: $p < 0.001$, Rayleigh test). However, there was only a modest increase of 17% in the entrainment strength (mean kappa) during the psychedelic state compared to baseline ($p = 0.034$, nested ANOVA). We found no significant difference between the 5-HT2A and NMDA conditions in terms of entrainment strength ($p = 0.53$, nested ANOVA).

the HFOs in the olfactory bulb had almost zero phase difference, while the HFOs in the ventral striatum led the olfactory bulb by about 0.5 radians. Orbitofrontal HFOs had similar phases, ranging between 0 and 0.5 radians in this example. These small —but often non-zero—phase differences were a general finding: In electrode pairs with detectable HFOs (median amplitude >5 μV, $n = 6237$), most pairs had a non-random phase difference (86% with kappa>1, compared to 71% on baseline; the median kappa increased by 47% compared to baseline, $p < 0.001$, Wilcoxon signed rank), and 95 % of those pairs had an absolute phase difference smaller than pi/4 (see the blue group in Fig. 5f and Fig. S11). When we compared the phase of the olfactory bulb to different structures, we saw small deviations from zero in all structures (ranging from 0.001 to 0.45 radians, corresponding to temporal delays of <1 ms; see Fig. 5g).

The observed near-zero phase lags are not consistent with a single source propagating synaptically or via volume conduction. An alternative explanation is that the HFOs are generated by a system of several self-sustaining but weakly interacting oscillators located in multiple structures. Such systems are known to have stable states with near-zero phase lags; similar to a standing wave[43]. To find support for the presence of local HFO generators outside the olfactory bulb, we looked for phase inversions in measurements from adjacent electrodes, since an inverted phase indicates that a local current dipole is present between the

electrodes. Such inversions were indeed found in about 2% of intrastructural electrode pairs (defined as an absolute phase difference larger than $3\pi/4$ and $\kappa>1$; see the red group in Fig. 5f) and in several structures, including the olfactory bulb, the olfactory cortex, the orbitofrontal cortex, the medial prefrontal cortex and the ventral striatum (Fig. 5h).

To further map the network of influences between structures, we calculated the Granger causality between structure pairs in the frequency domain. The Granger causality spectra often had clear peaks either in the HFO band or in the gamma band. During baseline, the peak of the Granger causality was mostly in the low gamma band, while it was mostly in the HFO band during the psychedelic state (Fig. 5i, j). The mean Granger causality in the HFO band (HFO frequency ±10 Hz) increased by 97% compared to baseline ($p < 0.001$, Wilcoxon rank sum), while it was not significantly changed in the gamma band (25–75 Hz; $p = 0.34$). A structure-by-structure analysis revealed that the Granger causality was clearly strongest from ventral striatum to mPFC (Fig. 5k).

In summary, the most specific neurophysiological correlate of psychedelic drug action was the enhancement of widespread phase-synchronized HFOs around 150 Hz. In the psychedelic state, we have identified a network consisting of the olfactory bulb, the olfactory cortex, the ventral striatum, the orbitofrontal cortex and the medial prefrontal cortex that are tightly coupled in the HFO band.

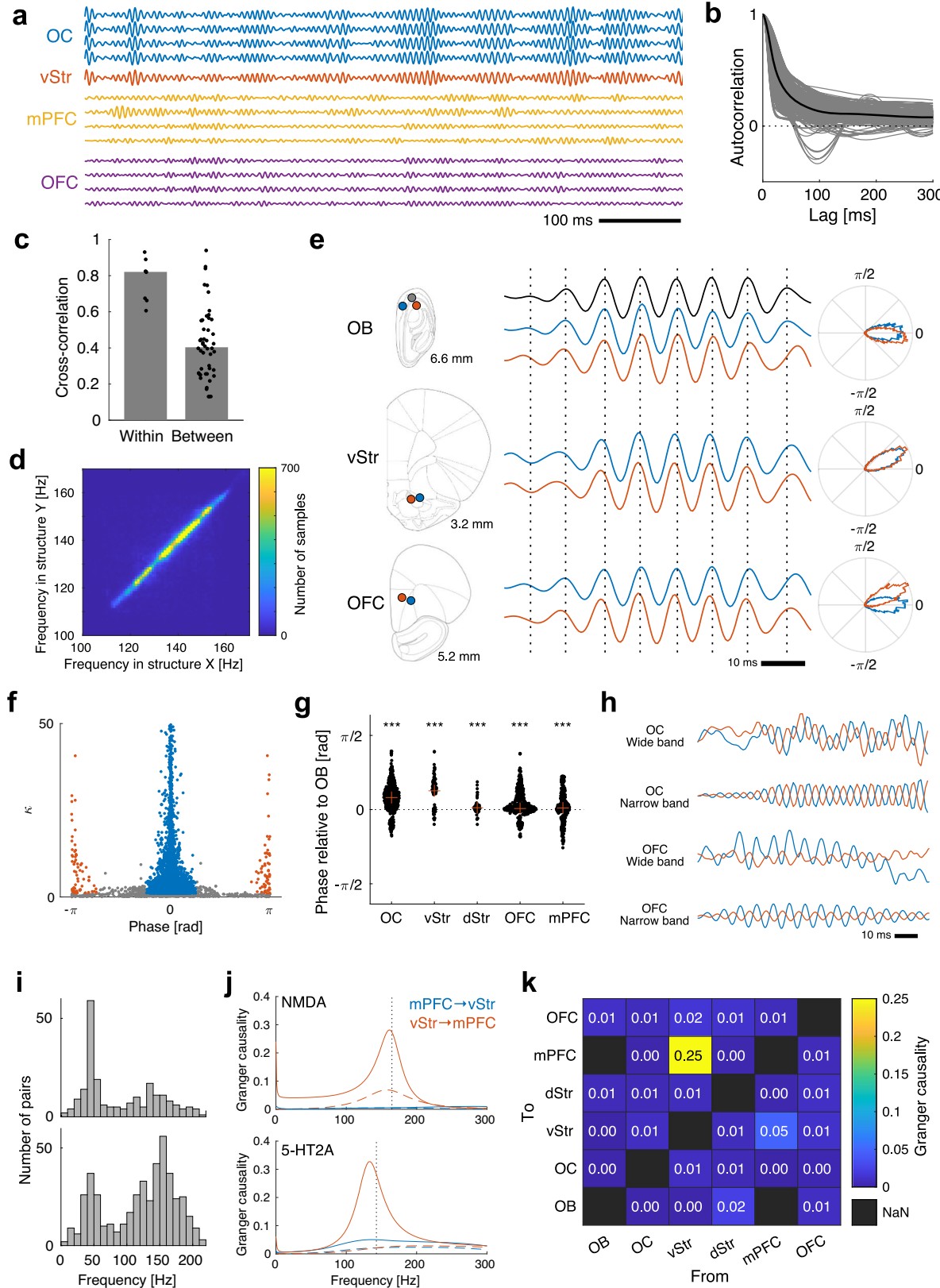

## Discussion

Several models propose explanations for how the known pharmacological effects on single neurons result in changes at the neuronal systems level, and how these in turn are related to subjective psychedelic experiences[2,37,44,45]. Some models explicitly state a direct link between firing rates and functional changes, like the thalamocortical gating model, in which increased mPFC excitability disinhibits the thalamus and reduces

**Fig. 5 HFOs are globally phase locked but have multiple sources. a** Example traces of bandpass filtered monopolar LFPs (110–190 Hz, zero-phase FIR) recorded simultaneously from the olfactory cortex (OC; blue), the ventral striatum (vStr; red), the medial prefrontal cortex (mPFC; yellow) and the orbitofrontal cortex (OFC; purple) during LSD treatment. Strong co-modulation of the HFO amplitude can be seen both within and between structures, as well as examples of independent modulation within and between structures. **b** Autocorrelograms of the instantaneous amplitude of all channels with clear HFOs during the psychedelic state (5-HT2A or NMDA). The black line is the average. Most autocorrelograms had a single clear peak (FWHM = 50±18 ms) consistent with spindle-like amplitude modulation with a spindle length of about 50 ms. **c** Cross-correlations of the instantaneous amplitude between all channel pairs with clear HFOs during the psychedelic state (5-HT2A or NMDA, $n = 1032$). Each dot shows the average cross-correlation for all pairs with channels in the same combination of anatomical structures. Pairs with both channels in the same anatomical structure ("Within") had higher cross-correlations on average than pairs with channels in different structures ("Between"). However, some pairs showed between-structure co-modulations that were similar in strength to the within-structure values. **d** 2D histogram showing the relationship between HFO frequencies in pairs of structures during the psychedelic state (5-HT2A or NMDA). Each data point comes from two simultaneously obtained spectra calculated from an 8 s time window. The high count on the diagonal shows that the frequency is very similar in all structures at any given time, despite a high degree of frequency modulation. **e** Example monopolar LFP traces showing a single HFO spindle recorded simultaneously from 7 electrodes in the olfactory bulb (OB; top), the ventral striatum (vStr; middle) and the orbitofrontal cortex (OFC; bottom) during treatment with LSD. Electrode positions are shown to the left. Vertical dashed lines are aligned to the peaks of the top OB trace (black) to facilitate comparisons of peak times between electrodes. Polar plots to the right show histograms of the phase difference of each electrode relative to the black OB electrode (based on the whole drug treatment period 30–60 min after injection). **f** Scatter plot showing mean phase differences and κ values for the phase difference distributions of each electrode pair ($n = 6237$). Most pairs had a non-random phase relationship (86% with κ > 1). Of those, 95% had a mean phase difference close to 0 ($|\varphi| < \frac{\pi}{4}$; blue dots) and 2% had an inverted phase ($|\varphi| > \frac{3}{4}\pi$; red dots). **g**. Swarm plot of phase differences relative to the olfactory bulb for all electrode pairs ($n = 1686$) grouped on structure. Each black dot is one electrode pair and red crosses indicate the median for the structure. A positive value means that the structure leads the olfactory bulb. Asterisks indicate that medians are significantly different from zero at the $p < 0.05$ (*), $p < 0.01$ (**) and $p < 0.001$ (***) levels (Wilcoxon signed rank). OB olfactory bulb, OC olfactory cortex, vStr ventral striatum, dStr dorsal striatum, OFC orbitofrontal cortex, mPFC medial prefrontal cortex. **h** Examples of phase inversion in monopolar LFP traces from two nearby electrodes in the olfactory cortex (OC) and two nearby electrodes in the orbitofrontal cortex (OFC) during ketamine treatment. This indicates the presence of local current dipoles between each electrode pair. Both raw LFP traces ("Wide") and bandpass filtered traces ("Narrow") are shown. Note that the OC pair is not recorded simultaneously with the OFC pair in this example. **i** Histograms of the frequency of the highest peak in the Granger causality spectra between pairs of bipolar measurements during baseline (top) and during the psychedelic state (5-HT2A or NMDA; bottom). Pairs were included if they came from structures with prevalent HFOs (OC, OFC, mPFC, vStr, and dStr) and if the peak value was larger than 0.2. During baseline, dominant peaks were most frequent in the classic gamma range around 40 Hz. During the psychedelic state, dominant peaks were instead most frequent in the HFO band around 150 Hz. **j** Example Granger causality spectra for one bipolar measurement in medial prefrontal cortex (mPFC) and one bipolar measurement in ventral striatum (vStr) during an NMDA antagonist experiment (ketamine; top) and a 5-HT2A experiment (LSD; bottom). The blue spectra show the causality of mPFC on vStr, while the red spectra show the causality in the opposite direction. Solid lines show the drug treated period and dashed lines show the corresponding baselines. The dotted vertical lines indicate the HFO frequency in the corresponding recording. **k** Median Granger causality values during the psychedelic state (5-HT2A or NMDA) calculated from the spectrum peak in the HFO band for all co-recorded structures with clear HFOs. Black squares indicate missing data.

its ability to gate sensory information[46]. Other models have indirectly linked the activation of certain cell populations to the disintegration of canonical network states, as seen for example in the reduced BOLD correlation between nodes in the default mode network[39]. Existing models have, however, highlighted apparently contradictory evidence and the field has suffered from an almost complete lack of in vivo electrophysiology data from awake animals acquired with an experimental design that could link the pharmacological effects on single cells with changes in information processing in the brain as a whole[47].

With simultaneous large-scale microwire recordings in multiple structures in freely behaving animals, we here showed that different classes of psychedelics affect firing rates differently, while they cause similar changes in population dynamics in the form of aberrantly strong HFOs. The HFOs were reliably detected in multiple structures despite using bipolar measurements that strongly attenuates distant sources. Together with the presence of phase reversals in several structures, this indicates that the HFOs were generated by local current dipoles in anatomically distinct regions (c.f. Olszewski et al.[48]). We also showed that the HFOs facilitate functional coupling between brain structures, both in terms of phase synchronization and Granger causality. This could have major effects on the exchange and integration of information across these neuronal systems. For example, similar types of excessive synchronization and coupling are associated with motor dysfunctions when they appear in the motor domain of the cortex-basal ganglia system of both rodents and humans[49].

To our knowledge, there is only one previous study on the effects of 5-HT2AR psychedelics on neuronal firing patterns in awake rats[15]. In that study, the authors recorded from the orbitofrontal and anterior cingulate cortices and found that DOI dose-dependently inhibited population activity. Our data showed similar inhibition in OFC and prelimbic mPFC (which is the area most closely related to the anterior cingulate in our dataset). In addition, we could show for the first time that 5-HT2AR psychedelics inhibit both the interneuron and principal cell populations, and that the inhibitory effect is not limited to the frontal cortex, but extends to all recorded regions, including the ventral striatum, the integrative (mediodorsal) thalamus and the temporal association area.

Our results cast doubts on models suggesting that specific changes in firing rates are directly linked to the psychedelic state, since the appearance of HFOs is largely independent of population firing rates. This is a surprising finding given the hypothesis that both 5-HT2AR and NMDAR psychedelics exert their effect via increased release of glutamate in corticolimbic circuits[11]. Intuitively, increased glutamate release should lead to increased excitation. However, biological neuronal networks have several homeostatic mechanisms to regulate the balance between excitation and inhibition, which makes it difficult to predict how changes in glutamate signaling affect the overall behavior of the network. Indeed, the seminal studies reporting increased AMPA-dependent excitatory currents in pyramidal cells did not observe simultaneous increases in firing rates of those neurons[13,50,51] and, more generally, it is well known that psychedelics are not pro-convulsant (see also[52]).

Perhaps it is fruitful to focus less on the excitatory role of glutamate, and more on how it may change the strength and

temporal dynamics of synaptic efficacy. Theoretical work has shown that the strength and dynamics of synaptic coupling play crucial roles in determining network behavior and, in particular, if a network has stable periodic states[53]. Intriguingly, both 5-HT2AR and NMDAR psychedelics decrease NMDA-mediated synaptic currents[54], while they increase AMPA-mediated currents (as mentioned above). However, AMPA channels have a much shorter deactivation time constant than NMDA channels (AMPA: 2–5 ms, NMDA: 50–100 ms), which will lead to dramatically faster temporal dynamics of the excitatory postsynaptic currents. This decreased "temporal smoothing" could in turn enable stable oscillatory states with a shorter period than would otherwise be possible.

While this might be a possible explanation for the appearance of local oscillatory states, the long-range HFO synchronization observed in the current study is still perplexing. Intuitively, it seems unlikely that such fast oscillations can synchronize across long distances considering the sizeable delays caused by the propagation of action potentials and the delayed activation of chemical synapses. On the other hand, gap junctions and ephaptic coupling could influence neighboring neurons almost instantaneously, but have very short range. However, mathematical analysis of idealized coupled oscillators has shown that stable synchronous states can exist with only local connectivity and even with delayed influences[43,55]. Interestingly, such systems often display a surprising complexity, where multiple stable synchronous states can co-exist and have different synchronization frequencies[56].

Taken together, the current study represents a first step towards bridging psychedelics-induced physiological phenomena at the single-cell level, via networks, to global brain states. Increasing our mechanistic understanding of how this class of substances induce psychedelic states will be essential to develop improved therapies for several neuropsychiatric conditions.

## Methods

**Animals**. Nine Sprague-Dawley rats (8 females, 1 male, 6-12 months old; Taconic, Denmark) were used in this study. Animals were kept in plastic cages on a 12:12-h light-dark cycle (lights on from 5:30 h to 17:30 h). All procedures complied with the ethical regulations and permits as approved in advance by the Malmö/Lund ethical committee of animal experiments.

**Construction of electrode arrays**. Microelectrode arrays with 128 wires were built and implanted as previously described[57], and allowed us to record local field potentials (LFP) and single units from up to 128 channels, for several weeks in freely moving animals. Briefly, formvar-insulated tungsten wires (33 and 50 μm diameter, California Fine Wire Co., CA) were arranged and cut to target up to 7 regions bilaterally: olfactory, orbitofrontal, sensorimotor, and prefrontal cortices, the hippocampus, ventral striatum and mediodorsal thalamus. The wires were soldered to a connector on a custom-made circuit board and the whole ensemble was fixated with UV curing.

**Implantation surgery**. Implantation surgeries were performed under fentanyl/medetomidine anesthesia (0.3/0.3 mg/kg, i.p.) and followed the procedures previously described[57]. Using a micromanipulator, the electrodes arrays were implanted in both hemispheres, and fixated with dental acrylic attached to 5–8 screws in the skull. A 200 μm thick silver wire was attached to three of these screws located on the occipital bone and was used as a ground connection from the animal to the recording system. After the implantation, the anesthesia was reversed by atipamezole hydrochloride (5 mg/kg, i.p.). Saline and the postoperative analgesic buprenorphine (0.5 mg/kg, subcutaneous injection) were administered. The animals were allowed to recover for at least one week, during which they were daily monitored and had their diet supplemented with a nutrient fortified water gel when needed (DietGel® Recovery, Karlslunde, Denmark).

**Verification of electrode positions**. Electrode positions were verified postmortem in 5 animals as follows: Animals were anesthetized with a lethal dose of sodium pentobarbital (100 mg/kg) and transcardially perfused with saline 0.9% (room temperature) followed by paraformaldehyde 4% (4 °C). Heads were stored in paraformaldehyde at 4 °C until scanned with computed tomography (CT) on a MILabs XUHR system (MILabs, Netherlands; 65 kV peak energy, 0.13 mA current,

90 ms exposure time, 200 microns Cu filter, 100 microns Al filter). The head was oriented so that the electrode wires were perpendicular to the photon beam to minimize artifacts. CT volumes were reconstructed with a voxel size of 20 microns and wire tips were identified semi-automatically in the CT volumes using custom-made software (https://github.com/NRC-Lund/ct-tools)[58]. The volumes were registered to an anatomical atlas[59] using lambda and bregma as landmarks, but a manual calibration was usually necessary to optimize the alignment between atlas and scan visually. The resulting affine transformation was used to calculate the atlas coordinates of the wire tips from the voxel coordinates (see Fig. 2a). Finally, the wire tips were assigned appropriate anatomical labels based on their location in the atlas. A total of 50 unique anatomical labels were attributed to the electrode tips in this way and they were further grouped into 10 broader regions based on assumed functional similarity (see Tables S1, S4).

**Pharmacological treatments**. To record the behavioral and electrophysiological effects of pharmacological treatments, animals were placed in a round open field arena and the implant was connected to the amplifier boards. After ~60 min of baseline recording, the animal was intraperitoneally injected with LSD (lysergic acid diethylamid, 0.3 mg/kg, Lipomed AG, Switzerland), DOI (2,5-dimethoxy-4-iodoamphetamine hydrochloride, 2 mg/kg, Lipomed AG, Switzerland), ketamine (Ketaminol, 25 - 50 mg/kg, Intervet AB, Sweden), PCP (phencyclidine hydrochloride, 5 mg/kg, Lipomed AG, Switzerland) or amphetamine (d-amphetamine sulfate, 4 mg/kg, Tocris, UK) and recorded for another 60-120 minutes. Data were averaged over -35 to -5 min for baseline measurements and 30 to 60 minutes for on-drug measurements (relative to drug injection). For each animal, the experiment was repeated with a different drug in a pseudorandomized order (in total 18 LSD, 9 DOI, 27 ketamine, 7 PCP, and 10 amphetamine experiments) after a washout period of at least 24 hours (the median washout period was 4 days; see Fig. 1a and Table S5).

**Signal acquisition**. Local field potential (LFP) and single unity activity were recorded with the Neuralynx multichannel recording system using a unity gain preamplifier (Neuralynx, MT, USA) or with the OpenEphys acquisition system[60] using 4 Intan RHD2132 amplifier boards with on-board AD-converters and accelerometers (Intan technologies, CA, USA).

For Neuralynx recordings, LFP signals were filtered between 0.1 and 300 Hz, and were digitized at 1017 Hz. Unit activities were filtered between 600 and 9000 Hz and digitized at 32 kHz. Thresholds for storage of spiking events in each channel were set to 2.5 SDs of the unfiltered signal.

For OpenEphys recordings, wideband signals were digitized and recorded at 30 kHz after bandpass filtering between 0.1 Hz and 10 kHz. LFPs were extracted offline by low pass filtering (8th order Butterworth at 500 Hz) and downsampling to 2000 Hz. Spike waveforms were extracted offline by thresholding at 2 SDs after bandpass filtering between 600 Hz and 9000 Hz (128th order FIR) and extracting 1 ms before and 2 ms after the threshold crossing event. The detector had a dead period of 1 ms. In addition, 3-axis accelerometer data was digitized and recorded at 30 kHz from all 4 amplifier boards.

Video was recorded at 25 fps with a camera placed above the open field arena. It was synchronized to the electrophysiology system using a Master-8 pulse generator (AMPI, Israel).

**Behavioral scoring**. Behavior was scored offline from the videos for 1 min every 10 min. Behaviors were scored from 0 to 3 depending on their prevalence (0 = not present, 1 = present for more than 5 seconds, 2 = present for more than 30 s, 3 = present continuously). The following behaviors were scored: being still, grooming, rearing, sniffing upwards, sniffing downwards, head-swaying, moving backwards, intermittent turning, unstableness, falling over, lying down and crawling. See Table S6 for more detailed definitions of the scored behaviors. Scores of head-sway, unstableness, falling over, lying down and crawling were averaged to an ataxia score. Similarly, scores of sniffing upwards, sniffing downwards and moving backwards were averaged to a stereotypy score[61].

**Video tracking**. Object tracking was performed using algorithms in Matlab adapted from[62]. Briefly, the foreground was separated from the static background using luminosity thresholding and foreground blobs were tracked between frames using a Kalman filter. The blob belonging to the animal (as opposed to the cable, for example) was identified based on shape parameters. The position of the animal was defined as the blob centroid.

Locomotion speed was calculated based on the translation of the blob centroid during a 1 s window. The distance traveled was calculated as the sum of the speed time series. Normally, rats prefer to stay along the arena walls and the time spent in the center was quantified and interpreted as a measure of disorientation or an increased drive to explore. The center area was defined as a circle with 2/3 the radius of the arena.

**Head-twitch response**. Head-twitch responses (HTR) were detected using the on-board accelerometers on the Intan RHD2132 amplifier boards that were attached to the dorsal side of the head via rigid adaptors. The mediolateral acceleration signal was downsampled to 200 Hz and bandpass filtered forwards and backwards with a finite impulse response filter (passband 8–32 Hz, filter order 100). A HTR index

was constructed by convoluting the absolute value of the filtered signal with a Gaussian window ($\sigma = 50$ ms). HTR events were extracted by detecting peaks in the HTR index that were higher than the threshold (0.4 g) and were separated by more than the detector dead time (1 s; see Fig. S12).

The method was validated with a manual inspection of the videos in 23 recordings. The validation resulted in a true positive rate of 95% with a false positive rate of 0.1 %. The method was implemented as Matlab code and is available at https://github.com/NRC-Lund/htrdetector.

Neuralynx recordings lacked accelerometer data and were analyzed manually from the videos.

**Spike sorting**. Extracted spikes were dejittered and clustered according to a hierarchical clustering scheme using a 2 ms refractory period[63]. Noise clusters were detected based on the normalized spike density during the refractory period and were removed if the density was larger than 1. Finally, all clusters were manually reviewed to determine if spike waveforms, firing rates, autocorrelation functions and interspike-interval distributions were physiologically plausible. About 12% of the clusters survived the manual review. We decided to always treat units from different recording sessions as independent units, despite the possibility that the same unit was recorded in multiple sessions.

Units were classified into putative cell types based on waveform features (valley width, peak width, and peak-to-valley time) as previously described[64]. The widths were defined as the full width at half maximum (FWHM). The classification was performed by fuzzy $k$-means clustering with probability of membership > 0.75, i.e., units with a probability of membership ≤ 0.75 were labeled as unclassified. Figure S5 and Table S7 show the clustering and summarizes the statistics of the waveform features of the cell types in all structures.

**Spike train analysis**. For firing rate analysis, spike times were binned in 10-second bins. Standardized rates were calculated by subtracting the average baseline rate and dividing with the standard deviation of the baseline rate. Spike entrainment to LFP oscillations was estimated by calculating the instantaneous phase of the bandpassed LFP for each spike and fitting a von Mises function to the resulting distribution using the *circ_vmpar* function in the CircStat toolbox[65]. To avoid the possibility that spiking affected the estimation of the LFP phase ("spike leakage"), the phase was calculated from all electrodes in the same anatomical structure as the neuron, but the electrode that contained the neuron was excluded.

**LFP power spectral densities**. To emphasize local sources of the measured electrical potential, bipolar LFP time series were computed from all unique pairs of electrodes from the same structure. For each of these time series, spectrograms were calculated with 50%-overlapping 8-s windows (0.12 Hz resolution) using the Irregularly Resampled AutoSpectral Analysis method (IRASA)[66]. IRASA separates the arrhythmic, so-called fractal component $S_{fractal}(f)$ from the power spectrum $S(f)$, and by normalizing the spectrum to the fractal component, it is possible to construct a power spectrum measure that emphasizes truly rhythmic activity:

$$S_{dB(fractal)}(f) = 10 \log \frac{S(f)}{S_{fractal}(f)} \quad (1)$$

The spectra $S_{db(fractal)}$ from individual electrode pairs were averaged structure-by-structure and fed into a peak detection algorithm that used nonlinear least-square fitting (Matlab *fit* function) to fit each spectrum to the following model:

$$y(f) = a_1 e^{-\left(\frac{f-a_2}{a_3}\right)^2} + a_4 f + a_5 \quad (2)$$

This allowed us to quantify peak height ($a_1$) and peak frequency ($a_2$) parametrically, but we also used this model as an oscillation detector by defining a threshold for the goodness-of-fit ($R^2$) and limits for the fitted parameters. Typical conditions for a positive HFO detection were $R^2 > 0.2$, $2 < a_1 < 100$ dB, $90 < a_2 < 170$ Hz, $1 < a_3 < 20$ Hz, $-1 < a_4 < 1$ and $-10 < a_5 < 10$ (see Fig. S8).

**Instantaneous phase and amplitude**. To quantify the instantaneous phase and amplitude of HFOs, monopolar LFP time series were bandpass filtered ±5 Hz around the median HFO frequency of each recording (as determined by the $a_2$ parameter above). We used a 64-order FIR filter (Matlab *fir1* function) backwards and forwards to ensure zero phase lag (Matlab *filtfilt* function). The bandpassed signal was Hilbert transformed into the complex-valued analytical signal $z(t)$ (Matlab *hilbert* function) and instantaneous phase and amplitude could then be calculated as $\varphi(t) = \arg z(t)$ and $r(t) = |z(t)|$.

Amplitude auto- and cross-correlations were calculated from the instantaneous amplitude time series in windows of 60 seconds and were then averaged across windows. For auto-correlations, only channels with a median amplitude above 15 μV were considered. To compare cross-correlations within and between structures, the height of the peak of the cross-correlogram was calculated for each electrode pair.

The mean phase difference between a pair of wires $i$ and $j$ was calculated as $\Delta\varphi_{ij} = \langle\varphi_i(t) - \varphi_j(t)\rangle$, where $\langle\rangle$ denotes the circular mean (function *circ_mean*, Matlab CircStat toolbox[65]. Similarly, the resultant vector length $r_{ij}$ was obtained using the function *circ_r* of the same toolbox. The mean phase difference between two brain regions was estimated by averaging all $\Delta\varphi_{ij}$. However, pairs with $r_{ij} < 0.5$ were excluded

from the average to ensure that only sufficiently stable phase difference estimates were used. Two wires were said to be phase inverted with respect to each other if $\left|\Delta\varphi_{ij}\right| > \frac{3}{4}\pi$ and if $\kappa > 1$, where $\kappa$ is the concentration parameter of the von Mises distribution.

**Granger causality**. Granger causality was calculated from bipolar LFP time series using the *one_bi_ga* function of the BSMART toolbox[67] with a 500 ms window length and a model order of 5. Typically, the Granger causality spectrum showed a clear peak in the gamma band or the HFO band. The frequency and amplitude of the peak were detected using the Matlab *findpeaks* function with default settings. Only the highest peak was analyzed further if multiple peaks were detected. The total Granger causality between regions was estimated as the median of the amplitude of the Granger causality peaks from all relevant wire pairs.

**Statistics and reproducibility**. Comparisons of behavioral measures were done with a nested ANOVA model using the Matlab *anovan* function[68] with factors State (baseline versus drug), Session and Animal. The factor Session was nested in Animal. Session and Animal were defined as random variables. A separate ANOVA was performed for each drug and for each evaluated parameter. The significance level was set to $\alpha = 0.05$.

To determine if the firing rate of a unit was significantly modulated compared to its own baseline, we performed a Wilcoxon rank sum test for equal medians (Matlab *ranksum* function, $\alpha = 0.01$) on spike counts binned in 10 s intervals. We used a binomial test (Matlab *binocdf* function) to determine if the number of significantly modulated cells in a population was higher or lower than chance ($\alpha = 0.05$). Standardized population rates were compared to baseline with a nested ANOVA model with factors State (baseline vs drug), Session, Animal, Time and Neuron (each sample consisting of the standardized rate from one time bin from one neuron). The factor Session was nested in Animal, while Time was nested in State and Neuron was nested in Session and Animal. Session, Animal and Neuron were defined as random variables and Time was defined as a continuous variable.

HFO amplitude and frequency were compared to baseline with a nested ANOVA model with factors State (baseline vs drug), Session, Animal, Hemisphere and Structure. The factor Session was nested in Animal. Session and Animal were defined as random variables. The model was identical for comparisons between drugs, except that State now had the levels "5HT2A" and "NMDA", and that Session was nested in both Animal and State.

To determine if a unit was significantly entrained to HFOs, we tested for non-uniformity of the phase distribution using a Reyleigh test (CircTest toolbox, *circ_rtest* function, $\alpha = 0.001$)[65].

Differences in cross-correlations and Granger causality were tested with the Wilcoxon rank sum test.

**Reporting summary**. Further information on research design is available in the Nature Portfolio Reporting Summary linked to this article.

## Data availability

The experimental data that support the findings in this study are available in the EBRAINS repository with the identifier https://doi.org/10.25493/MV16-7TC[69]. The source data for the graphs are available in Supplementary Data 1.

## Code availability

Data was analyzed using standard methods implemented as Matlab code. The code is available upon reasonable request. Code for detecting head-twitch responses is available at https://github.com/NRC-Lund/htrdetector. Code for localizing electrode tips in CT images is available at https://github.com/NRC-Lund/ct-tools.

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

## Acknowledgements

Lund University Bioimaging Centre (LBIC), Lund University, is gratefully acknowledged for providing experimental resources. The IRASA computations were enabled by resources provided by the Swedish National Infrastructure for Computing (SNIC) at LUNARC.

The study was supported by grants from BABEL (Erasmus Mundus), Crafoord Foundation, Insamlingsstiftelserna, Kempe Foundation, Barncancerfonden, Kocks Foundation, Kungliga Fysiografiska Sällskapet, Magnus Bergvall Foundation, Olle Engkvist Foundation, Oskarfonden, Parkinsonfonden, Petrus och Augusta Hedlunds Stiftelse, Promobilia, Segerfalk Foundation, Sigur & Elsa Goljes Minne Foundation, Sven-Olof Jansons livsverk, Svenska Sällskapet för Medicinsk Forskning (SSMF), Thurings Foundation, Umeå Universitet, Vetenskapsrådet (Grant #2021-01769, #2018-02717 and #2016-07213) Wenner-Gren Foundation and Åhlén Foundations.

## Author contributions

I.B., P.P., and P.H. designed experiments. I.B. and S.B. performed experiments assisted by E.W. and J.W. P.H. analyzed the data. E.W. performed additional behavioral analysis. J.W. contributed to spike analysis. P.H. wrote the paper with contributions from I.B., S.B., and P.P.

## Funding

## Competing interests

The authors declare no competing interests.
