## [Peer Review File · Communications Biology]

REVIEWERS' COMMENTS:

Reviewer #1 (Remarks to the Author):

The authors have substantially revised their previous version and have introduced a new main figure and several supplementary figures.

As I mentioned in my previous review my main issue was novelty. Some of the figures, especially Figure 1 (behavior) and 3 (HFO) are largely predicted from what is already known in the field. The authors have done a good job with the behavioral characterization (a lot is in the supplementary data). The behavioral findings are not novel, but do serve to show they drug effects work in the expected manner, so in this sense are justified. In terms of the oscillatory work, mainly HFO, I would agree with the authors that their current work does indeed go beyond what has been done previously, in terms of recordings from multiple brain regions simultaneously. The unit work, again from multi-structures is also an advance.

There are other weaknesses in the current study, in particular, the short washout period, which the authors have explained was due to their ethics permission. Notwithstanding, the results seem robust enough and the differences obtained appear consistent, so short washout does not seem to be a serious issue in this study (but please see point 6, below).

Strengths: This is a thorough and careful study examining the effect of psychoactive compounds on LFP and unit activity across many brain regions. Most previous work has either addressed LFPs or units, but not together. Recordings from so many areas is a technical achievement. It does advance earlier studies by the authors' multi-structure approach. Overall, I think this work is a useful advance to the field of psychopharmacology and should be a highly cited paper.

Weaknesses: The work although largely descriptive does not address mechanisms. Even though there is a lot of data, some data are confirmatory and it is not immediately obvious what the breakthrough result is. I think the authors should more clearly frame this, and the implications, in the abstract and discussion.

Specific Comments:

1. In the results section the authors write: "Importantly, in the further analyses of links between behaviors and neuronal activity, we also confirmed that psychedelic-specific changes in neuronal activity cannot be explained trivially by changes in the concomitantly displayed motor behavior". Where are the analyses? All I could find was the the data shown in Fig 3A from an example rat with the speed underneath.
2. Figure 3 I believe is bipolar, so please add this to the caption somewhere. In C, SM cortex should be defined, and also the abbreviations used in E.
3. Figure 5. In this figure can the authors state on the figure if the waveforms in A, E, and H are mono or bipolar. Also they say "recorded synchronously" but better would be "recorded simultaneously". H is also after LSD, right?
4. S6 There is an asterisk for the interneurons in the temporal association area but no bar is visible.
5. S10 Sensorimotor striatum, I think the authors mean dorsal striatum.
6. S12 The authors mention show phase data for HFO and spiking at baseline. Considering that these rats are not naïve and some would have been injected with a drugs that increases HFO power only 24 hours previously this effect may be related to the drug rather than true baseline (drug free) effects. I think the only way the authors can address this is if they analyse the baseline data for the first

experiment. As only this way can longer lasting drug effects be ruled out. If the authors can't do this because there is not enough spike data in the baseline fragment then washout/multiple injection effects should be mentioned in this context. A color scheme would be helpful for S12 as well.